# High-Dimensional Sparse Linear Bandits

**Botao Hao**
Deepmind
haobotao000@gmail.com

**Tor Lattimore**
Deepmind
lattimore@google.com

**Mengdi Wang**
Department of Electrical Engineering
Princeton University
mengdiw@princeton.edu

## Abstract

Stochastic linear bandits with high-dimensional sparse features are a practical model for a variety of domains, including personalized medicine and online advertising [Bastani and Bayati, 2020]. We derive a novel $\Omega(n^{2/3})$ dimension-free minimax regret lower bound for sparse linear bandits in the data-poor regime where the horizon is smaller than the ambient dimension and where the feature vectors admit a well-conditioned exploration distribution. This is complemented by a nearly matching upper bound for an explore-then-commit algorithm showing that that $\Theta(n^{2/3})$ is the optimal rate in the data-poor regime. The results complement existing bounds for the data-rich regime and provide another example where carefully balancing the trade-off between information and regret is necessary. Finally, we prove a dimension-free $\mathcal{O}(\sqrt{n})$ regret upper bound under an additional assumption on the magnitude of the signal for relevant features.

## 1 Introduction

Stochastic linear bandits generalize the standard reward model for multi-armed bandits by associating each action with a feature vector and assuming the mean reward is the inner product between the feature vector and an unknown parameter vector [Auer, 2002, Dani et al., 2008, Rusmevichientong and Tsitsiklis, 2010, Chu et al., 2011, Abbasi-Yadkori et al., 2011].

In most practical applications, there are many candidate features but no clear indication about which are relevant. Therefore, it is crucial to consider stochastic linear bandits in the high-dimensional regime but with low-dimensional structure, captured here by the notion of sparsity. Previous work on sparse linear bandits has mostly focused on the data-rich regime, where the time horizon is larger than the ambient dimension [Abbasi-Yadkori et al., 2012, Carpentier and Munos, 2012, Wang et al., 2018, Kim and Paik, 2019, Bastani and Bayati, 2020]. The reason for studying the data-rich regime is partly justified by minimax lower bounds showing that for smaller time horizons the regret is linear in the worst case.

Minimax bounds, however, do not tell the whole story. A crude maximisation over all environments hides much of the rich structure of linear bandits with sparsity. We study sparse linear bandits in the high-dimensional regime when the ambient dimension is much larger than the time horizon. In order to sidestep existing lower bounds, we refine the minimax notion by introducing a dependence in our bounds on the minimum eigenvalue of a suitable exploration distribution over the actions. Similar

Table 1: Comparisons with existing results on regret upper bounds and lower bounds for sparse linear bandits. Here, $s$ is the sparsity, $d$ is the feature dimension, $n$ is the number of rounds, $K$ is the number of arms, $C_{\min}$ is the minimum eigenvalue of the data matrix for an exploration distribution (3.1) and $\tau$ is a problem-dependent parameter that may have a complicated form and vary across different literature.

| Upper Bound | Regret | Assumptions | Regime |
|---|---|---|---|
| Abbasi-Yadkori et al. [2012] | $\mathcal{O}(\sqrt{sdn})$ | none | rich |
| Sivakumar et al. [2020] | $\mathcal{O}(\sqrt{sdn})$ | adver. + Gaussian noise | rich |
| Bastani and Bayati [2020] | $\mathcal{O}(\tau K s^2 (\log(n))^2)$ | compatibility condition | rich |
| Wang et al. [2018] | $\mathcal{O}(\tau K s^3 \log(n))$ | compatibility condition | rich |
| Kim and Paik [2019] | $\mathcal{O}(\tau s \sqrt{n})$ | compatibility condition | rich |
| Lattimore et al. [2015] | $\mathcal{O}(s\sqrt{n})$ | action set is hypercube | rich |
| This paper (Thm. 4.2) | $\mathcal{O}(C_{\min}^{-2/3} s^{2/3} n^{2/3})$ | action set spans $\mathbb{R}^d$ | poor |
| This paper (Thm. 5.2) | $\mathcal{O}(C_{\min}^{-1/2} \sqrt{sn})$ | action set spans $\mathbb{R}^d$ + mini. signal | rich |
| **Lower Bound** | | | |
| Multi-task bandits[1] | $\Omega(\sqrt{sdn})$ | N.A. | rich |
| This paper (Thm. 3.3) | $\Omega(C_{\min}^{-1/3} s^{1/3} n^{2/3})$ | N.A. | poor |

quantities appear already in the vast literature on high-dimensional statistics [Bühlmann and Van De Geer, 2011, Wainwright, 2019].

**Contributions**   Our first result is a lower bound showing that $\Omega(n^{2/3})$ regret is generally unavoidable when the dimension is large, even if the action set admits an exploration policy for which the minimum eigenvalue of the associated data matrix is large. The lower bound is complemented by an explore-the-sparsity-then-commit algorithm that first solves a convex optimization problem to find the most informative design in the exploration stage. The algorithm then explores for a number of rounds by sampling from the design distribution and uses Lasso [Tibshirani, 1996] to estimate the unknown parameters. Finally, it greedily chooses the action that maximizes the reward given the estimated parameters. We derive an $\mathcal{O}(n^{2/3})$ dimension-free regret that depends instead on the minimum eigenvalue of the covariance matrix associated with the exploration distribution. Our last result is a post-model selection linear bandits algorithm that invokes phase-elimination algorithm [Lattimore et al., 2020] to the model selected by the first-step regularized estimator. Under a sufficient condition on the minimum signal of the feature covariates, we prove that a dimension-free $\mathcal{O}(\sqrt{n})$ regret is achievable, even if the data is scarce.

The analysis reveals a rich structure that has much in common with partial monitoring, where $\Theta(n^{2/3})$ regret occurs naturally in settings for which some actions are costly but highly informative [Bartók et al., 2014]. A similar phenomenon appears here when the dimension is large relative to the horizon. There is an interesting transition as the horizon grows, since $\mathcal{O}(\sqrt{dn})$ regret is optimal in the data rich regime.

**Related work**   Most previous work is focused on the data-rich regime. For an arbitrary action set, Abbasi-Yadkori et al. [2012] proposed an online-to-confidence-set conversion approach that achieves a $\mathcal{O}(\sqrt{sdn})$ regret upper bound, where $s$ is a known upper bound on the sparsity. The algorithm is generally not computationally efficient, which is believed to be unavoidable. Additionally, a $\Omega(\sqrt{sdn})$ regret lower bound for data-rich regime was established in Section 24.3 of Lattimore and Szepesvári [2020], which means polynomial dependence on $d$ is generally not avoidable without additional assumptions.

For this reason, it recently became popular to study the contextual setting, where the action set changes from round to round and to careful assumptions are made on the context distribution. The

assumptions are chosen so that techniques from high-dimensional statistics can be borrowed. Suppose $\tau$ is a problem-dependent parameter that may have a complicated form and varies across different literature. Kim and Paik [2019] developed a doubly-robust Lasso bandit approach with an $\mathcal{O}(\tau s\sqrt{n})$ upper bound but required the average of the feature vectors for each arm satisfies the compatibility condition [Bühlmann and Van De Geer, 2011]. Bastani and Bayati [2020] and Wang et al. [2018] considered a multi-parameter setting (each arm has its own underlying parameter) and assumed the distribution of contexts satisfies a variant of the compatibility condition as well as other separation conditions. Bastani and Bayati [2020] derived a $\mathcal{O}(\tau K s^2(\log(n))^2)$ upper bound and was sharpen to $\mathcal{O}(\tau K s^2 \log(n))$ by Wang et al. [2018], where $K$ is the number of arms. Although those results are dimension-free, they require strong assumptions on the context distribution that are hard to verify in practice. As a result, the aforementioned regret bounds involved complicated problem-dependent parameters that may be very large when the assumptions fail to hold.

Another thread of the literature is to consider specific action sets. Lattimore et al. [2015] proposed a selective explore-then-commit algorithm that only works when the action set is exactly the binary hypercube. They derived an optimal $\mathcal{O}(s\sqrt{n})$ upper bound as well as an optimal gap-dependent bound. Sivakumar et al. [2020] assumed the action set is generated adversarially but perturbed artificially by some standard Gaussian noise. They proposed a structured greedy algorithm to achieve an $\mathcal{O}(s\sqrt{n})$ upper bound. Deshpande and Montanari [2012] study the data-poor regime in a Bayesian setting but did not consider sparsity. Carpentier and Munos [2012] considered a special case where the action set is the unit sphere and the noise is vector-valued so that the noise becomes smaller as the dimension grows. We summarize the comparisons in Table 1.

## 2 Problem setting

In the beginning, the agent receives a compact action set $\mathcal{A} \subseteq \mathbb{R}^d$, where $d$ may be larger than the number of rounds $n$. At each round $t$, the agent chooses an action $A_t \in \mathcal{A}$ and receives a reward

$$Y_t = \langle A_t, \theta \rangle + \eta_t, \tag{2.1}$$

where $(\eta_t)_{t=1}^n$ is a sequence of independent standard Gaussian random variables and $\theta \in \mathbb{R}^d$ is an unknown parameter vector. We make the mild boundedness assumption that for all $x \in \mathcal{A}$, $\|x\|_\infty \le 1$. The parameter vector $\theta$ is assumed to be $s$-sparse:

$$\|\theta\|_0 = \sum_{j=1}^d \mathbb{1}\{\theta_j \ne 0\} \le s.$$

The Gaussian assumption can be relaxed to conditional sub-Gaussian assumption for the regret upper bound, but is necessary for the regret lower bound. The performance metric is the cumulative expected regret, which measures the difference between the expected cumulative reward collected by the omniscient policy that knows $\theta$ and that of the learner. The optimal action is $x^* = \mathrm{argmax}_{x \in \mathcal{A}} \langle x, \theta \rangle$ and the regret of the agent when facing the bandit determined by $\theta$ is

$$R_\theta(n) = \mathbb{E}\left[\sum_{t=1}^n \langle x^*, \theta \rangle - \sum_{t=1}^n Y_t\right],$$

where the expectation is over the interaction sequence induced by the agent and environment. Our primary focus is on finite-time bounds in the data-poor regime where $d \ge n$.

**Notation** Let $[n] = \{1, 2, \ldots, n\}$. For a vector $x$ and positive semidefinite matrix $A$, we let $\|x\|_A = \sqrt{x^\top A x}$ be the weighted $\ell_2$-norm and $\sigma_{\min}(A), \sigma_{\max}(A)$ be the minimum eigenvalue and maximum eigenvalue of $A$, respectively. The cardinality of a set $\mathcal{A}$ is denoted by $|\mathcal{A}|$. The support of a vector $x$, $\mathrm{supp}(x)$, is the set of indices $i$ such that $x_i \ne 0$. And $\mathbb{1}\{\cdot\}$ is an indicator function. The suboptimality gap of action $x \in \mathcal{A}$ is $\Delta_x = \langle x^*, \theta \rangle - \langle x, \theta \rangle$ and the minimum gap is $\Delta_{\min} = \min\{\Delta_x : x \in \mathcal{A}, \Delta_x > 0\}$.

# 3   Minimax lower bound

As promised, we start by proving a kind of minimax regret lower. We first define a quantity that measures the degree to which there exist good exploration distributions over the actions.

**Definition 3.1.** Let $\mathcal{P}(\mathcal{A})$ be the space of probability measures over $\mathcal{A}$ with the Borel $\sigma$-algebra and define

$$C_{\min}(\mathcal{A}) = \max_{\mu \in \mathcal{P}(\mathcal{A})} \sigma_{\min}\Big( \mathbb{E}_{A \sim \mu}\big[ AA^\top \big] \Big).$$

**Remark 3.2.** $C_{\min}(\mathcal{A}) > 0$ if and only if $\mathcal{A}$ spans $\mathbb{R}^d$. Two illustrative examples are the hypercube and probability simplex. Sampling uniformly from the corners of each set shows that $C_{\min}(\mathcal{A}) \geq 1$ for the former and $C_{\min}(\mathcal{A}) \geq 1/d$ for the latter.

The next theorem is a kind of minimax lower bound for sparse linear bandits. The key steps of the proof follow, with details and technical lemmas deferred to Appendix B.

**Theorem 3.3.** Consider the sparse linear bandits described in Eq. (2.1). Then for any policy $\pi$ there exists an action set $\mathcal{A}$ with $C_{\min}(\mathcal{A}) > 0$ and $s$-sparse parameter $\theta \in \mathbb{R}^d$ such that

$$R_\theta(n) \geq \frac{\exp(-4)}{4} \min\Big( C_{\min}^{-\frac{1}{3}}(\mathcal{A}) s^{\frac{1}{3}} n^{\frac{2}{3}}, \sqrt{dn} \Big). \tag{3.1}$$

Theorem 3.3 holds for any data regime and suggests an intriguing transition between $n^{2/3}$ and $n^{1/2}$ regret, depending on the relation between the horizon and the dimension. When $d > n^{1/3} s^{2/3}$ the bound is $\Omega(n^{2/3})$, which is independent of the dimension. On the other hand, when $d \leq n^{1/3} s^{2/3}$, we recover the standard $\Omega(\sqrt{sdn})$ dimension-dependent lower bound up to a $\sqrt{s}$-factor. In Section 4, we prove that the $\Omega(n^{2/3})$ minimax lower bound is tight by presenting a nearly matching upper bound in the data-poor regime.

**Remark 3.4.** Theorem 3.3 has a worst-case flavor. For each algorithm we construct a problem instance with the given dimension, sparsity and value of $C_{\min}$ for which the stated regret bound holds. The main property of this type of hard instance is that it should include a informative but high-regret action set such that the learning algorithm should balance the trade-off between information and regret. This leaves the possibility for others to create minimax lower bound for their own problem.

*Proof of Theorem 3.3.* The proof uses the standard information-theoretic machinery, but with a novel construction and KL divergence calculation.

**Step 1: construct a hard instance.** We first construct a low regret action set $\mathcal{S}$ and an informative action set $\mathcal{H}$ as follows:

$$
\begin{aligned}
\mathcal{S} &= \Big\{ x \in \mathbb{R}^d \Big| x_j \in \{-1, 0, 1\} \text{ for } j \in [d-1], \|x\|_1 = s-1, x_d = 0 \Big\}, \\
\mathcal{H} &= \Big\{ x \in \mathbb{R}^d \Big| x_j \in \{-\kappa, \kappa\} \text{ for } j \in [d-1], x_d = 1 \Big\},
\end{aligned}
\tag{3.2}
$$

where $0 < \kappa \leq 1$ is a constant. The action set is the union $\mathcal{A} = \mathcal{S} \cup \mathcal{H}$ and let

$$\theta = \big( \underbrace{\varepsilon, \dots, \varepsilon}_{s-1}, 0, \dots, 0, -1 \big),$$

where $\varepsilon > 0$ is a small constant to be tuned later. Because $\theta_d = -1$, actions in $\mathcal{H}$ are associated with a large regret. On the other hand, actions in $\mathcal{H}$ are also highly informative, which hints towards an interesting tradeoff between regret and information. Note that $\mathcal{H}$ is nearly the whole binary hypercube, while actions in $\mathcal{S}$ are $(s-1)$-sparse. The optimal action is in the action set $\mathcal{A}$:

$$x^* = \operatorname*{argmax}_{x \in \mathcal{A}} \langle x, \theta \rangle = \big( \underbrace{1, \cdots, 1}_{s-1}, 0, \dots, 0 \big) \in \mathcal{A}. \tag{3.3}$$

**Step 2: construct an alternative bandit**. The second step is to construct an alternative bandit $\widetilde{\theta}$ that is hard to distinguish from $\theta$ and for which the optimal action for $\theta$ is suboptimal for $\widetilde{\theta}$ and vice versa.

Denote $\mathbb{P}_\theta$ and $\mathbb{P}_{\widetilde{\theta}}$ as the measures on the sequence of outcomes $(A_1, Y_1, \ldots, A_n, Y_n)$ induced by the interaction between a fixed bandit algorithm and the bandits determined by $\theta$ and $\widetilde{\theta}$ respectively. Let $\mathbb{E}_\theta, \mathbb{E}_{\widetilde{\theta}}$ be the corresponding expectation operators. We denote a set $\mathcal{S}'$ as

$$\mathcal{S}' = \Big\{ x \in \mathbb{R}^d \Big| x_j \in \{-1, 0, 1\} \text{ for } j \in \{s, s+1, \ldots, d-1\}, \tag{3.4}$$
$$x_j = 0 \text{ for } j = \{1, \ldots, s-1, d\}, \|x\|_1 = s-1 \Big\}.$$

Clearly, $\mathcal{S}'$ is a subset of $\mathcal{S}$ and for any $x \in \mathcal{S}'$, its support has no overlap with $\{1, \ldots, s-1\}$. Then we denote

$$\widetilde{x} = \underset{x \in \mathcal{S}'}{\operatorname{argmin}} \, \mathbb{E}_\theta \left[ \sum_{t=1}^n \langle A_t, x \rangle^2 \right], \tag{3.5}$$

and construct the alternative bandit $\widetilde{\theta}$ as

$$\widetilde{\theta} = \theta + 2\varepsilon\widetilde{x}. \tag{3.6}$$

Note that $\widetilde{\theta}$ is $(2s-1)$-sparse since $\widetilde{x}$ belongs to $\mathcal{S}'$ that is a $(s-1)$-sparse set. This design guarantees the optimal arm $x^*$ in bandit $\theta$ is suboptimal in alternative bandit $\widetilde{\theta}$ and the suboptimality gap for $x^*$ in bandit $\widetilde{\theta}$ is $\max_{x \in \mathcal{A}} \langle x - x^*, \widetilde{\theta} \rangle = (s-1)\varepsilon$. Define an event

$$\mathcal{D} = \left\{ \sum_{t=1}^n \mathbb{1}(A_t \in \mathcal{S}) \sum_{j=1}^{s-1} A_{tj} \leq \frac{n(s-1)}{2} \right\}.$$

The next claim shows that when $\mathcal{D}$ occurs, the regret is large in bandit $\theta$, while if it does not occur, then the regret is large in bandit $\widetilde{\theta}$. The detailed proof is deferred to Appendix B.1.

**Claim 3.5.** Regret lower bounds with respect to event $\mathcal{D}$:

$$R_\theta(n) \geq \frac{n(s-1)\varepsilon}{2} \mathbb{P}_\theta(\mathcal{D}) \qquad \text{and} \qquad R_{\widetilde{\theta}}(n) \geq \frac{n(s-1)\varepsilon}{2} \mathbb{P}_{\widetilde{\theta}}(\mathcal{D}^c).$$

By the Bretagnolle–Huber inequality (Lemma C.1 in the appendix),

$$R_\theta(n) + R_{\widetilde{\theta}}(n) \geq \frac{n(s-1)\varepsilon}{2} \Big( \mathbb{P}_\theta(\mathcal{D}) + \mathbb{P}_{\widetilde{\theta}}(\mathcal{D}^c) \Big) \geq \frac{n(s-1)\varepsilon}{4} \exp\Big( -\operatorname{KL}\big(\mathbb{P}_\theta, \mathbb{P}_{\widetilde{\theta}}\big) \Big),$$

where $\operatorname{KL}(\mathbb{P}_\theta, \mathbb{P}_{\widetilde{\theta}})$ is the KL divergence between probability measures $\mathbb{P}_\theta$ and $\mathbb{P}_{\widetilde{\theta}}$.

**Step 3: calculating the KL divergence.** We make use of the following bound on the KL divergence between $\mathbb{P}_\theta$ and $\mathbb{P}_{\widetilde{\theta}}$, which formalises the intuitive notion of information. When the KL divergence is small, the algorithm is unable to distinguish the two environments. The detailed proof is deferred to Appendix B.2.

**Claim 3.6.** Define $T_n(\mathcal{H}) = \sum_{t=1}^n \mathbb{1}(A_t \in \mathcal{H})$. The KL divergence between $\mathbb{P}_\theta$ and $\mathbb{P}_{\widetilde{\theta}}$ is upper bounded by

$$\operatorname{KL}(\mathbb{P}_\theta, \mathbb{P}_{\widetilde{\theta}}) \leq 2\varepsilon^2 \left( \frac{n(s-1)^2}{d} + \kappa^2(s-1)\mathbb{E}_\theta[T_n(\mathcal{H})] \right). \tag{3.7}$$

The first term in the right-hand side of the bound is the contribution from actions in the low-regret action set $\mathcal{S}$, while the second term is due to actions in $\mathcal{H}$. The fact that actions in $\mathcal{S}$ are not very informative is captured by the presence of the dimension in the denominator of the first term. When $d$ is very large, the algorithm simply does not gain much information by playing actions in $\mathcal{S}$. When $T_n(\mathcal{H}) < 1/(\kappa^2(s-1)\varepsilon^2)$, it is easy to see

$$R_\theta(n) + R_{\widetilde{\theta}}(n) \geq \frac{n(s-1)\varepsilon}{4} \exp\left( -\frac{2n\varepsilon^2(s-1)^2}{d} \right) \exp(-2). \tag{3.8}$$

On the other hand, when $T_n(\mathcal{H}) > 1/(\kappa^2\varepsilon^2(s-1))$, we have

$$R_\theta(n) \geq \mathbb{E}_\theta[T_n(\mathcal{H})] \min_{x \in \mathcal{H}} \Delta_x \geq \frac{1}{\kappa^2\varepsilon^2(s-1)} + \frac{1-\kappa}{\kappa^2\varepsilon}, \tag{3.9}$$

since $\min_{x \in \mathcal{H}} \Delta_x = 1 + (s-1)\varepsilon(1-\kappa)$ from the definition of $\mathcal{H}$ and $\theta$.

**Step 4: conclusion.** Combining the above two cases together, we have

$$R_\theta(n) + R_{\widetilde{\theta}}(n) \geq \min\left(\left(\frac{ns\varepsilon}{4}\right)\exp\left(-\frac{2\varepsilon^2 s^2 n}{d}\right)\exp(-2), \frac{1}{\kappa^2\varepsilon^2 s} + \frac{1-\kappa}{\kappa^2\varepsilon}\right), \qquad (3.10)$$

where we replaced $s - 1$ by $s$ in the final result for notational simplicity. Consider a sampling distribution $\mu$ that uniformly samples actions from $\mathcal{H}$. A simple calculation shows that $C_{\min}(\mathcal{A}) \geq C_{\min}(\mathcal{H}) \geq \kappa^2 > 0$. This is due to

$$\sigma_{\min}\left(\sum_{x \in \mathcal{H}} \mu(x) x x^\top\right) = \sigma_{\min}\left(\mathbb{E}_{X \sim \mu}[XX^\top]\right) = \kappa^2,$$

where each coordinate of the random vector $X \in \mathbb{R}^d$ is sampled independently uniformly from $\{-1, 1\}$. In the data poor regime when $d \geq n^{1/3} s^{2/3}$, we choose $\varepsilon = \kappa^{-2/3} s^{-2/3} n^{-1/3}$ such that

$$\max(R_\theta(n), R_{\widetilde{\theta}}(n)) \geq R_\theta(n) + R_{\widetilde{\theta}}(n)$$
$$\geq \frac{\exp(-4)}{4}\kappa^{-\frac{2}{3}} s^{\frac{1}{3}} n^{\frac{2}{3}} \geq \frac{\exp(-4)}{4} C_{\min}^{-\frac{1}{3}}(\mathcal{A}) s^{\frac{1}{3}} n^{\frac{2}{3}}.$$

Finally, in the data rich regime when $d < n^{1/3} s^{2/3}$ we choose $\varepsilon = \sqrt{d/(ns^2)}$ such that the exponential term is a constant, and then

$$\max(R_\theta(n), R_{\widetilde{\theta}}(n)) \geq R_\theta(n) + R_{\widetilde{\theta}}(n) \geq \frac{\exp(-4)}{4}\sqrt{dn}. \qquad \qquad \square$$

## 4  Matching upper bound

We now propose a simple algorithm based on the explore-then-commit paradigm[2] and show that the minimax lower bound in Eq. (3.1) is more or less achievable. As one might guess, the algorithm has two stages. First it solves the following optimization problem to find the most informative design:

$$\max_{\mu \in \mathcal{P}(\mathcal{A})} \sigma_{\min}\left(\int_{x \in \mathcal{A}} x x^\top d\mu(x)\right). \qquad (4.1)$$

In the exploration stage, the agent samples its actions from $\widehat{\mu}$ for $n_1$ rounds, collecting a data-set $\{(A_1, Y_1), \ldots, (A_{n_1}, Y_{n_1})\}$. The agent uses the data collecting in the exploration stage to compute the Lasso estimator $\widehat{\theta}_{n_1}$. In the commit stage, the agent executes the greedy action for the rest $n - n_1$ rounds. The detailed algorithm is summarized in Algorithm 1.

**Remark 4.1.** The minimum eigenvalue is concave [Boyd et al., 2004], which means that the solution to (4.1) can be approximated efficiently using standard tools such as CVXPY [Diamond and Boyd, 2016].

The following theorem states a regret upper bound for Algorithm 1. The proof is deferred to Appendix B.3.

**Theorem 4.2.** Consider the sparse linear bandits described in Eq. (2.1) and assume the action set $\mathcal{A}$ spans $\mathbb{R}^d$. Suppose $R_{\max}$ is an upper bound of maximum expected reward such that $\max_{x \in \mathcal{A}}\langle x, \theta \rangle \leq R_{\max}$. In Algorithm 1, we choose

$$n_1 = n^{2/3}(s^2 \log(2d))^{1/3} R_{\max}^{-2/3}(2/C_{\min}^2(\mathcal{A}))^{1/3}, \qquad (4.3)$$

and $\lambda_1 = 4\sqrt{\log(d)/n_1}$. Then the following regret upper bound holds,

$$R_\theta(n) \leq (2\log(2d) R_{\max})^{\frac{1}{3}} C_{\min}^{-\frac{2}{3}}(\mathcal{A}) s^{\frac{2}{3}} n^{\frac{2}{3}} + 3n R_{\max} \exp(-c_1 n_1). \qquad (4.4)$$

**Algorithm 1** Explore the sparsity then commit (ESTC)

---

1: **Input:** time horizon $n$, action set $\mathcal{A}$, exploration length $n_1$, regularization parameter $\lambda_1$;
2: Solve the optimization problem in Eq. (4.1) and denote the solution as $\widehat{\mu}$.
3: **for** $t = 1, \cdots, n_1$ **do**
4:   Independently pull arm $A_t$ according to $\widehat{\mu}$ and receive a reward: $Y_t = \langle A_t, \theta \rangle + \eta_t$.
5: **end for**
6: Calculate the Lasso estimator [Tibshirani, 1996]:

$$\widehat{\theta}_{n_1} = \operatorname*{argmin}_{\theta \in \mathbb{R}^d} \left( \frac{1}{n_1} \sum_{t=1}^{n_1} \left( Y_t - \langle A_t, \theta \rangle \right)^2 + \lambda_1 \|\theta\|_1 \right). \tag{4.2}$$

7: **for** $t = n_1 + 1$ to $n$ **do**
8:   Take greedy actions $A_t = \operatorname{argmin}_{x \in \mathcal{A}} \langle \widehat{\theta}_{n_1}, x \rangle$.
9: **end for**

---

Together with the minimax lower bound in Theorem 3.3, we can argue that ESTC algorithm is minimax optimal in time horizon $n$ in the data-poor regime.

**Remark 4.3.** The regret upper bound Eq. (4.4) may still depend on $d$ because $1/C_{\min}(\mathcal{A})$ could be as large as $d$. Indeed, if the action set is the standard basis vectors, then the problem reduces to the standard multi-armed bandit for which the minimax regret is $\Theta(\sqrt{dn})$, even with sparsity. If we restrict our attention to the class of action set such that $1/C_{\min}(\mathcal{A})$ is dimension-free, then we have a dimension-free upper bound.

**Remark 4.4.** Another notion frequently appearing in high-dimensional statistics is the restricted eigenvalue condition. Demanding a lower bound on the restricted eigenvalue is weaker than the minimum eigenvalue, which can lead to stronger results. As it happens, however, the two coincide in the lower bound construction. The upper bound may also be sharpened, but the resulting optimization problem would (a) depend on the sparsity $s$ and (b) the objective would have a complicated structure for which an efficient algorithm is not yet apparent.

**Remark 4.5.** There is still a $(s/C_{\min}(\mathcal{A}))^{1/3}$ gap between the lower bound (Eq. (3.1)) and upper bound (Eq. (4.4)) ignoring logarithmic factor. We conjecture that the use of $\ell_1/\ell_\infty$ inequality when proving Theorem 4.2 is quite conservative. Specifically, we bound the following using the $\ell_1$-norm bound of Lasso (see Eq. (B.15) in the Appendix B.3 for details),

$$\langle \theta - \widehat{\theta}_{n_1}, x^* - A_t \rangle \leq \left\| \theta - \widehat{\theta}_{n_1} \right\|_1 \left\| x^* - A_t \right\|_\infty \lesssim \sqrt{\frac{s^2 \log(d)}{n_1}}.$$

The first inequality ignores the sign information of $\widehat{\theta}_{n_1}$ and the correlation between $x^* - A_t$ and $\widehat{\theta}_{n_1}$. A similar phenomenon has been observed by Javanmard et al. [2018] and resolved by means of a delicate leave-one-out analysis to decouple the correlation. An interesting question is whether or not a similar technique could be used in our case to improve the above bound to $\sqrt{s \log(d)/(n_1)}$, closing the gap between regret upper bound and lower bound. On the other hand, surprisingly, even in the classical statistical settings there are still gaps between upper and lower bounds in terms of $C_{\min}(\mathcal{A})$ [Raskutti et al., 2011]. We speculate that the upper bound may be improvable, though at present we do not know how to do it.

**Remark 4.6.** The algorithm uses knowledge of the sparsity to tune the length of exploration in Eq. (4.3). When the sparsity is not known, the length of exploration can be set to $n_1 = n^{2/3}$. The price is an additional factor of $\mathcal{O}(s^{1/3})$ to regret. This is an advantage relative to the algorithm by Abbasi-Yadkori et al. [2012], for which knowledge of the sparsity is apparently essential for constructing the confidence set.

**Remark 4.7.** We do not expect explicit optimism-based algorithms [Dani et al., 2008, Rusmevichientong and Tsitsiklis, 2010, Chu et al., 2011, Abbasi-Yadkori et al., 2011] or implicit ones, such as Thompson sampling [Agrawal and Goyal, 2013], to achieve the minimax lower bound in the data-poor

regime. The reason is that the optimism principle does not balance the trade-off between information and regret, a phenomenon that has been observed before in linear and structured bandits [Lattimore and Szepesvari, 2017, Combes et al., 2017, Hao et al., 2020].

# 5 Improved upper bound

In this section, we show that under additional minimum signal condition, the restricted phase elimination algorithm can achieve a sharper $\mathcal{O}(\sqrt{sn})$ regret upper bound.

The algorithm shares similar idea with Carpentier and Munos [2012] that includes feature selection step and restricted linear bandits step. In the feature selection step, the agent pulls a certain number of rounds $n_2$ following $\widehat{\mu}$ as in (4.1). Then Lasso is used to conduct the feature selection. Based on the support Lasso selects, the algorithm invokes phased elimination algorithm for linear bandits [Lattimore et al., 2020] on the selected support.

---

**Algorithm 2** Restricted phase elimination

1: **Input:** time horizon $n$, action set $\mathcal{A}$, exploration length $n_2$, regularization parameter $\lambda_2$;
2: Solve the optimization problem Eq. (4.1) and denote the solution as $\widehat{\mu}$.
3: **for** $t = 1, \cdots, n_2$ **do**
4:     Independently pull arm $A_t$ according to $\widehat{\mu}$ and receive a reward: $Y_t = \langle A_t, \theta \rangle + \eta_t$.
5: **end for**
6: Calculate the Lasso estimator $\widehat{\theta}_{n_2}$ as in Eq. (4.2) with $\lambda_2$.
7: Identify the support: $\widehat{S} = \text{supp}(\widehat{\theta}_{n_2})$.
8: **for** $t = n_2 + 1$ to $n$ **do**
9:     Invoke phased elimination algorithm for linear bandits on $\widehat{S}$.
10: **end for**

---

**Condition 5.1** (Minimum signal). We assume the minimum non-zero component of $\theta$ satisfies:

$$\min_{j \in \text{supp}(\theta)} |\theta_j| > \frac{1}{C_{\min}(\mathcal{A})} \sqrt{\frac{4s \log(d)}{n}}.$$

**Theorem 5.2.** Consider the sparse linear bandits described in Eq. (2.1). We assume the action set $\mathcal{A}$ spans $\mathbb{R}^d$ as well as $|\mathcal{A}| = K < \infty$ and suppose Condition 5.1 holds. Let $n_2 = C_1 s \log(d)$ for a suitable large constant $C_1$ and choose $\lambda_2 = 4\sqrt{\log(d)/n_2}$. Denote $\phi_{\max} = \sigma_{\max}(\sum_{t=1}^{n_2} A_t A_t^\top / n_2)$. Then the following regret upper bound of Algorithm 2 holds,

$$R_\theta(n) \leq C \Big( s \log(d) + \sqrt{\frac{9\phi_{\max} \log(Kn)}{C_{\min}(\mathcal{A})}} \sqrt{sn} \Big), \tag{5.1}$$

for universal constant $C > 0$.

The proof is deferred to Appendix B.4. It utilizes the sparsity and variable screening property of Lasso. More precisely, under minimum signal condition, the Lasso estimator can identify all the important covariates, i.e., $\text{supp}(\widehat{\theta}_{n_1}) \supseteq \text{supp}(\theta)$. And the model Lasso selected is sufficiently sparse, i.e. $|\text{supp}(\widehat{\theta}_{n_1})| \lesssim s$. Therefore, it is enough to query linear bandits algorithm on $\text{supp}(\widehat{\theta}_{n_1})$.

**Remark 5.3.** It is possible to remove the dependency of $\phi_{\max}$ in the Eq. (5.1) using more dedicated analysis, using theorem 3 in Belloni et al. [2013]. The reason we choose a phase elimination type algorithm is that it has the optimal regret guarantee when the size of action set is moderately large. When the action set has an infinite number of actions, we could switch to the linear UCB algorithm [Abbasi-Yadkori et al., 2011] or appeal to a discretisation argument.

# 6 Experiment

We compare ESTC (our algorithm) with LinUCB [Abbasi-Yadkori et al., 2011] and doubly-robust (DR) lasso bandits [Kim and Paik, 2019]. For ESTC, we use the theoretically suggested length of

exploration stage. For LinUCB, we use the theoretically suggested confidence interval. For DR-lasso, we use the code made available by the authors on-line.

- **Case 1: linear contextual bandits.** We use the setting in Section 5 of Kim and Paik [2019] with $N = 20$ arms, dimension $d = 100$, sparsity $s = 5$. At round $t$, we generate the action set from $N(0_N, V)$, where $V_{ii} = 1$ and $V_{ik} = \rho^2$ for every $i \neq k$. Larger $\rho$ corresponds to high correlation setting that is more favorable to DR-lasso. The noise is from $N(0, 1)$ and $\|\theta\|_0 = s$.
- **Case 2: hard problem instance.** Consider the hard problem instance in the proof of minimax lower bound (Theorem 3.3), including an informative action set and an uninformative action set. Since the size of action set constructed in the hard problem instance grows exponentially with $d$, we uniformly randomly sample 500 actions from the full informative action set and 200 from uninformative action set.

**Conclusion:** The experiments confirm our theoretical findings. Although our theory focuses on the fixed action set setting, ESTC works well in the contextual setting. DR-lasso bandits heavily rely on context distribution assumption and almost fail for the hard instance. LinUCB suffers in the data-poor regime since it ignores the sparsity information.

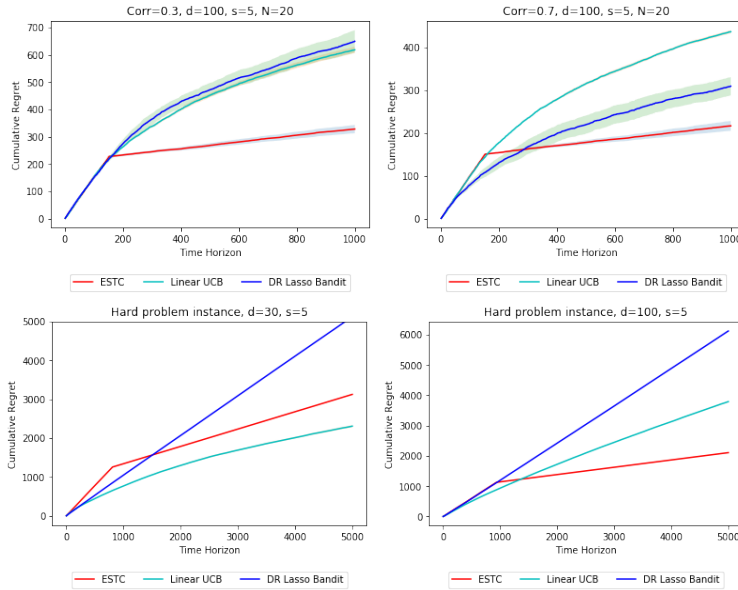

Figure 1: The top two figures are for Case 1 and the bottom two figures are for Case 2.

# 7 Discussion

In this paper, we provide a thorough investigation of high-dimensional sparse linear bandits, and show that $\Theta(n^{2/3})$ is the optimal rate in the data-poor regime. Our work leaves many open problems on how the shape of action set affects the regret that reveals the subtle trade-off between information and regret. For instance, it is unclear how the regret lower bound depends on $C_{\min}(\mathcal{A})$ in the data-rich regime and if $C_{\min}(\mathcal{A})$ is the best quantity to describe the shape of action set $\mathcal{A}$.

In another hand, the ESTC algorithm can only achieve optimal regret bound in data poor regime and becomes suboptimal in the data rich regime. It is interesting to have an algorithm to achieve optimal regrets in "best of two worlds". Information-direct sampling [Russo and Van Roy, 2014] might be a good candidate since it delicately balances the trade-off between information and regret which is necessary in the sparse linear bandits.

**Broader Impact**    We believe that presented research should be categorized as basic research and we are not targeting any specific application area. Theorems may inspire new algorithms and theoretical investigation. The algorithms presented here can be used for many different applications and a particular use may have both positive or negative impacts. We are not aware of any immediate short term negative implications of this research and we believe that a broader impact statement is not required for this paper.

## Acknowledgments and Disclosure of Funding

Mengdi Wang gratefully acknowledges funding from the U.S. National Science Foundation (NSF) grant CMMI1653435, Air Force Office of Scientific Research (AFOSR) grant FA9550-19-1-020, and C3.ai DTI.

## Footnotes

[1] Section 24.3 of Lattimore and Szepesvári [2020]

[2]Explore-then-commit template is also considered in other works [Deshmukh et al., 2018] but both the exploration and exploitation stages are very different. Deshmukh et al. [2018] considers simple regret minimization while we focus on cumulative regret minimization.

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
