[Supplementary Material]

In Appendix A, we review some statistical results for sparse linear regression. In Appendix B, we provide the proof of main theorems as well as main claims. In Appendix C, we include some supporting lemma for the sake of completeness.

## A  Sparse linear regression

We review some classical results in sparse linear regression. Consider the following sparse linear regression model:

$$y_i = \langle x_i, \theta^* \rangle + \epsilon_i, i = 1, \ldots, n, \tag{A.1}$$

where $\theta^* \in \mathbb{R}^d$ and $\|\theta^*\|_0 = s \leq d$ and the noise $\{\epsilon_i\}_{i=1}^n$ independently follows a zero-mean, $\sigma$-sub-Gaussian distribution. Let the design matrix be $X = (x_1, \ldots, x_n)^\top \in \mathbb{R}^{n \times d}$. Define the Lasso estimator as follows:

$$\widehat{\theta}_n = \operatorname*{argmin}_\theta \Big( \frac{1}{n} \sum_{i=1}^n (y_i - \langle x_i, \theta \rangle)^2 + \lambda \|\theta\|_1 \Big).$$

**Condition A.1** (Restricted eigenvalues). Define the cone:

$$\mathbb{C}(S) := \{\Delta \in \mathbb{R}^d | \|\Delta_{S^c}\|_1 \leq 3\|\Delta_S\|_1\},$$

where $S$ is the support set of $\theta^*$. Then there exists some positive constant $\kappa$ such that the design matrix $X \in \mathbb{R}^{n \times d}$ satisfied the condition

$$\frac{\|X\theta\|_2^2}{n} \geq \kappa \|\theta\|_2^2,$$

for all $\theta \in \mathbb{C}(S)$.

**Condition A.2** (Column normalized). Using $X_j \in \mathbb{R}^n$ to denote the $j$-th column of $X$, we say that $X$ is column-normalized if for all $j = 1, 2, \ldots, d$,

$$\frac{\|X_j\|_2}{\sqrt{n}} \leq 1.$$

**Theorem A.3.** Consider an $s$-sparse linear regression and assume design matrix $X \in \mathbb{R}^{n \times d}$ satisfies the RE condition (Condition A.1) and the column normalization condition (Condition (A.2)). Given the Lasso estimator with regularization parameter $\lambda_n = 4\sigma\sqrt{\log(d)/n}$, then with probability at least $1 - \delta$,

- the estimation error under $\ell_1$-norm (Theorem 7.13 in Wainwright [2019]) of any optimal solution $\widehat{\theta}_n$ satisfies

$$\big\|\widehat{\theta}_n - \theta^*\big\|_1 \leq \frac{\sigma s}{\kappa} \sqrt{\frac{2\log(2d/\delta)}{n}};$$

- the mean square prediction error (Theorem 7.20 in Wainwright [2019]) of any optimal solution $\widehat{\theta}_n$ satisfies

$$\frac{1}{n} \sum_{i=1}^n \big(x_i^\top(\widehat{\theta}_n - \theta)\big)^2 \leq \frac{9}{\kappa} \frac{s\log(d/\delta)}{n}.$$

## B  Proofs of main theorems and claims

### B.1  Proof of Claim 3.5

We first prove the first part. By standard calculations, we have

$$R_\theta(n) = \mathbb{E}_\theta\Big[\sum_{t=1}^n \langle x^*, \theta \rangle\Big] - \mathbb{E}_\theta\Big[\sum_{t=1}^n \langle A_t, \theta \rangle\Big]$$

$$= \mathbb{E}_\theta\Big[n(s-1)\varepsilon - \sum_{t=1}^n \mathbb{1}(A_t \in \mathcal{H})\langle A_t, \theta \rangle - \sum_{t=1}^n \mathbb{1}(A_t \in \mathcal{S})\langle A_t, \theta \rangle\Big],$$

where the last equation is from the definition of $x^*$ in Eq. (3.3). From the definition of $\mathcal{H}$ in Eq. (3.2), the following holds for small enough $\varepsilon$,

$$\sum_{t=1}^{n} \mathbb{1}(A_t \in \mathcal{H})\langle A_t, \theta \rangle \leq T_n(\mathcal{H})(\kappa(s-1)\varepsilon - 1) \leq 0, \tag{B.1}$$

where $T_n(\mathcal{H}) = \sum_{t=1}^{n} \mathbb{1}(A_t \in \mathcal{H})$. Since $\langle A_t, \theta \rangle = \sum_{j=1}^{s} A_{tj}\varepsilon$ for $A_t \in \mathcal{S}$, then it holds that

$$
\begin{aligned}
R_\theta(n) &\geq \mathbb{E}_\theta\Big[n(s-1)\varepsilon - \sum_{t=1}^{n} \mathbb{1}(A_t \in \mathcal{S}) \sum_{j=1}^{s-1} A_{tj}\varepsilon\Big] \\
&\geq \mathbb{E}_\theta\Big[\Big(n(s-1)\varepsilon - \sum_{t=1}^{n} \mathbb{1}(A_t \in \mathcal{S}) \sum_{j=1}^{s-1} A_{tj}\varepsilon\Big) \mathbb{1}(\mathcal{D})\Big] \\
&\geq \Big(n(s-1)\varepsilon - \frac{n(s-1)\varepsilon}{2}\Big)\mathbb{P}_\theta(\mathcal{D}) \\
&= \frac{n(s-1)\varepsilon}{2}\mathbb{P}_\theta(\mathcal{D}).
\end{aligned}
\tag{B.2}
$$

Second, we derive a regret lower bound of alternative bandit $\widetilde{\theta}$. Denote $\widetilde{x}^*$ as the optimal arm of bandit $\widetilde{\theta}$. By a similar decomposition in Eq. (B.2),

$$
\begin{aligned}
R_{\widetilde{\theta}}(n) &= \mathbb{E}_{\widetilde{\theta}}\Big[\sum_{t=1}^{n}\langle \widetilde{x}^*, \widetilde{\theta} \rangle\Big] - \mathbb{E}_{\widetilde{\theta}}\Big[\sum_{t=1}^{n}\langle A_t, \widetilde{\theta} \rangle\Big] \\
&= \mathbb{E}_{\widetilde{\theta}}\Big[2n(s-1)\varepsilon - \sum_{t=1}^{n} \mathbb{1}(A_t \in \mathcal{H})\langle A_t, \widetilde{\theta} \rangle - \sum_{t=1}^{n} \mathbb{1}(A_t \in \mathcal{S})\langle A_t, \widetilde{\theta} \rangle\Big] \\
&\geq \mathbb{E}_{\widetilde{\theta}}\Big[2n(s-1)\varepsilon - \sum_{t=1}^{n} \mathbb{1}(A_t \in \mathcal{S})\langle A_t, \widetilde{\theta} \rangle\Big].
\end{aligned}
\tag{B.3}
$$

where the inequality comes similarly in Eq. (B.1) to show $\sum_{t=1}^{n} \mathbb{1}(A_t \in \mathcal{H})\langle A_t, \widetilde{\theta} \rangle \leq 0$. Next, we will find an upper bound for $\sum_{t=1}^{n} \mathbb{1}(A_t \in \mathcal{S})\langle A_t, \widetilde{\theta} \rangle$. From the definition of $\widetilde{\theta}$ in Eq. (3.6),

$$
\begin{aligned}
\sum_{t=1}^{n} \mathbb{1}(A_t \in \mathcal{S})\langle A_t, \widetilde{\theta} \rangle &= \sum_{t=1}^{n} \mathbb{1}(A_t \in \mathcal{S})\langle A_t, \theta + 2\varepsilon\widetilde{x} \rangle \\
&= \sum_{t=1}^{n} \mathbb{1}(A_t \in \mathcal{S})\langle A_t, \theta \rangle + 2\varepsilon \sum_{t=1}^{n} \mathbb{1}(A_t \in \mathcal{S})\langle A_t, \widetilde{x} \rangle \\
&\leq \sum_{t=1}^{n} \mathbb{1}(A_t \in \mathcal{S})\langle A_t, \theta \rangle + 2\varepsilon \sum_{t=1}^{n} \mathbb{1}(A_t \in \mathcal{S}) \sum_{j \in \mathrm{supp}(\widetilde{x})} |A_{tj}|,
\end{aligned}
\tag{B.4}
$$

where the last inequality is from the definition of $\widetilde{x}$ in Eq. (3.5). To bound the first term, we have

$$
\begin{aligned}
\sum_{t=1}^{n} \mathbb{1}(A_t \in \mathcal{S})\langle A_t, \theta \rangle &= \sum_{t=1}^{n} \mathbb{1}(A_t \in \mathcal{S}) \sum_{j=1}^{s-1} A_{tj}\varepsilon \\
&\leq \varepsilon \sum_{t=1}^{n} \mathbb{1}(A_t \in \mathcal{S}) \sum_{j=1}^{s-1} |A_{tj}|.
\end{aligned}
\tag{B.5}
$$

If all the actions $A_t$ come from $\mathcal{S}$ which is a $(s-1)$-sparse set, we have

$$\sum_{t=1}^{n}\sum_{j=1}^{d} |A_{tj}| = (s-1)n,$$

which implies

$$\sum_{t=1}^{n} \mathbb{1}(A_t \in \mathcal{S}) \Big( \sum_{j=1}^{s-1} |A_{tj}| + \sum_{j \in \mathrm{supp}(\widetilde{x})} |A_{tj}| \Big) \leq \sum_{t=1}^{n} \mathbb{1}(A_t \in \mathcal{S}) \sum_{j=1}^{d} |A_{tj}| \leq (s-1)n,$$

$$\sum_{t=1}^{n} \mathbb{1}(A_t \in \mathcal{S}) \sum_{j=1}^{s-1} |A_{tj}| \leq (s-1)n - \sum_{t=1}^{n} \mathbb{1}(A_t \in \mathcal{S}) \sum_{j \in \mathrm{supp}(\widetilde{x})} |A_{tj}|. \tag{B.6}$$

Combining with Eq. (B.5),

$$\sum_{t=1}^{n} \mathbb{1}(A_t \in \mathcal{S}) \langle A_t, \theta \rangle \leq \varepsilon \Big( (s-1)n - \sum_{t=1}^{n} \mathbb{1}(A_t \in \mathcal{S}) \sum_{j \in \mathrm{supp}(\widetilde{x})} |A_{tj}| \Big)$$

Plugging the above bound into Eq. (B.4), it holds that

$$\sum_{t=1}^{n} \mathbb{1}(A_t \in \mathcal{S}) \langle A_t, \widetilde{\theta} \rangle \leq \varepsilon(s-1)n + \varepsilon \sum_{t=1}^{n} \mathbb{1}(A_t \in \mathcal{S}) \sum_{j \in \mathrm{supp}(\widetilde{x})} |A_{tj}|. \tag{B.7}$$

When the event $\mathcal{D}^c$ (the complement event of $\mathcal{D}$) happen, we have

$$\sum_{t=1}^{n} \mathbb{1}(A_t \in \mathcal{S}) \sum_{j=1}^{s-1} |A_{tj}| \geq \sum_{t=1}^{n} \mathbb{1}(A_t \in \mathcal{S}) \sum_{j=1}^{s-1} A_{tj} \geq \frac{n(s-1)}{2}.$$

Combining with Eq. (B.6), we have under event $\mathcal{D}^c$,

$$\sum_{t=1}^{n} \mathbb{1}(A_t \in \mathcal{S}) \sum_{j \in \mathrm{supp}(\widetilde{x})} |A_{tj}| \leq \frac{n(s-1)}{2}. \tag{B.8}$$

Putting Eqs. (B.3), (B.7), (B.8) together, it holds that

$$R_{\widetilde{\theta}}(n) \geq \frac{n(s-1)\varepsilon}{2} \mathbb{P}_{\widetilde{\theta}}(\mathcal{D}^c). \tag{B.9}$$

This ends the proof.

## B.2  Proof of Claim 3.6

From the divergence decomposition lemma (Lemma C.2 in the appendix), we have

$$\mathrm{KL}\big(\mathbb{P}_\theta, \mathbb{P}_{\widetilde{\theta}}\big) = \frac{1}{2} \mathbb{E}_\theta \Big[ \sum_{t=1}^{n} \langle A_t, \theta - \widetilde{\theta} \rangle^2 \Big]$$

$$= 2\varepsilon^2 \mathbb{E}_\theta \Big[ \sum_{t=1}^{n} \langle A_t, \widetilde{x} \rangle^2 \Big].$$

To prove the claim, we use a simple argument "minimum is always smaller than the average". We decompose the following summation over action set $\mathcal{S}'$ defined in Eq. (3.4),

$$\sum_{x \in \mathcal{S}'} \sum_{t=1}^{n} \langle A_t, x \rangle^2 = \sum_{x \in \mathcal{S}'} \sum_{t=1}^{n} \Big( \sum_{j=1}^{d} x_j A_{tj} \Big)^2$$

$$= \sum_{x \in \mathcal{S}'} \sum_{t=1}^{n} \Big( \sum_{j=1}^{d} (x_j A_{tj})^2 + 2 \sum_{i<j} x_i x_j A_{ti} A_{tj} \Big).$$

We bound the above two terms separately.

1. To bound the first term, we observe that

$$\sum_{x \in \mathcal{S}'} \sum_{t=1}^{n} \sum_{j=1}^{d} \left( x_j A_{tj} \right)^2$$

$$= \sum_{x \in \mathcal{S}'} \sum_{t=1}^{n} \mathbb{1}(A_t \in \mathcal{S}) \sum_{j=1}^{d} |x_j A_{tj}| + \sum_{x \in \mathcal{S}'} \sum_{t=1}^{n} \mathbb{1}(A_t \in \mathcal{H}) \sum_{j=1}^{d} (x_j A_{tj})^2, \qquad \text{(B.10)}$$

since both $x_j, A_{tj}$ can only take $-1, 0, +1$ if $A_t \in \mathcal{S}$. If all the $A_t$ come from $\mathcal{S}$, we have

$$\sum_{t=1}^{n} \sum_{j=1}^{d} |A_{tj}| = (s-1)n.$$

This implies

$$\sum_{t=1}^{n} \mathbb{1}(A_t \in \mathcal{S}) \sum_{j=1}^{d} |A_{tj}| \le (s-1)n.$$

Since $x \in \mathcal{S}'$ that is $(s-1)$-sparse, we have $\sum_{j=1}^{d} |x_j A_{tj}| \le s - 1$. Therefore, we have

$$\sum_{x \in \mathcal{S}'} \sum_{t=1}^{n} \mathbb{1}(A_t \in \mathcal{S}) \sum_{j=1}^{d} |x_j A_{tj}| \le (s-1)n \binom{d-s-1}{s-2}. \qquad \text{(B.11)}$$

In addition, since the action in $\mathcal{S}'$ is $s - 1$-sparse and has 0 at its last coordinate, we have

$$\sum_{x \in \mathcal{S}'} \sum_{t=1}^{n} \mathbb{1}(A_t \in \mathcal{H}) \sum_{j=1}^{d} (x_j A_{tj})^2 \le \kappa^2 |\mathcal{S}'| T_n(\mathcal{H})(s-1). \qquad \text{(B.12)}$$

Putting Eqs. (B.10), (B.11) and (B.12) together,

$$\sum_{x \in \mathcal{S}'} \sum_{t=1}^{n} \sum_{j=1}^{d} \left( x_j A_{tj} \right)^2 \le (s-1)n \binom{d-s-1}{s-2} + \kappa^2 |\mathcal{S}'| T_n(\mathcal{H})(s-1). \qquad \text{(B.13)}$$

2. To bound the second term, we observe

$$\sum_{x \in \mathcal{S}'} \sum_{t=1}^{n} 2 \sum_{i<j} x_i x_j A_{ti} A_{tj} = 2 \sum_{t=1}^{n} \sum_{i<j} \sum_{x \in \mathcal{S}'} x_i x_j A_{ti} A_{tj}.$$

From the definition of $\mathcal{S}'$, $x_i x_j$ can only take values of $\{1 * 1, 1 * -1, -1 * 1, -1 * -1, 0\}$. This symmetry implies

$$\sum_{x \in \mathcal{S}'} x_i x_j A_{ti} A_{tj} = 0,$$

which implies

$$\sum_{x \in \mathcal{S}'} \sum_{t=1}^{n} 2 \sum_{i<j} x_i x_j A_{ti} A_{tj} = 0. \qquad \text{(B.14)}$$

Combining Eqs. (B.13) and (B.14) together, we have

$$\sum_{x \in \mathcal{S}'} \sum_{t=1}^{n} \langle A_t, x \rangle^2 = \sum_{x \in \mathcal{S}'} \sum_{t=1}^{n} \sum_{j=1}^{d} |x_j A_{tj}|$$

$$\le (s-1)n \binom{d-s-1}{s-2} + \kappa^2 |\mathcal{S}'| T_n(\mathcal{H})(s-1).$$

Therefore, we use the fact that the minimum of $n$ points is always smaller than its average,

$$
\begin{aligned}
\mathbb{E}_\theta\Big[\sum_{t=1}^n \langle A_t, \widetilde{x}\rangle^2\Big] &= \min_{x\in\mathcal{S}'} \mathbb{E}_\theta\Big[\sum_{t=1}^n \langle A_t, x\rangle^2\Big] \\
&\leq \frac{1}{|\mathcal{S}'|} \sum_{x\in\mathcal{S}'} \mathbb{E}_\theta\Big[\sum_{t=1}^n \langle A_t, x\rangle^2\Big] \\
&= \mathbb{E}_\theta\Big[\frac{1}{|\mathcal{S}'|} \sum_{x\in\mathcal{S}'} \sum_{t=1}^n \langle A_t, x\rangle^2\Big] \\
&\leq \frac{(s-1)n\binom{d-s-1}{s-2} + \mathbb{E}_\theta[T_n(\mathcal{H})](s-1)\binom{d-s}{s-1}}{\binom{d-s}{s-1}} \\
&\leq \frac{(s-1)^2 n}{d} + \kappa^2 \mathbb{E}_\theta[T_n(\mathcal{H})](s-1).
\end{aligned}
$$

This ends the proof of the claim of Eq. (3.7).

## B.3 Proof of Theorem 4.2: regret upper bound

**Step 1: regret decomposition.** Suppose $R_{\max}$ is an upper bound of maximum expected reward such that $\max_{x\in\mathcal{A}}\langle x, \theta\rangle \leq R_{\max}$. We decompose the regret of ESTC as follows:

$$
\begin{aligned}
R_\theta(n) &= \mathbb{E}_\theta\Big[\sum_{t=1}^n \langle \theta, x^* - A_t\rangle\Big] \\
&= \mathbb{E}_\theta\Big[\sum_{t=1}^{n_1} \langle \theta, x^* - A_t\rangle + \sum_{t=n_1+1}^n \langle \theta, x^* - A_t\rangle\Big] \\
&\leq \mathbb{E}_\theta\Big[2n_1 R_{\max} + \sum_{t=n_1+1}^n \langle \theta - \widehat{\theta}_{n_1}, x^* - A_t\rangle + \sum_{t=n_1+1}^n \langle \widehat{\theta}_{n_1}, x^* - A_t\rangle\Big].
\end{aligned}
$$

Since we take greedy actions when $t \geq n_1 + 1$, it holds that $\langle x^*, \widehat{\theta}_{n_1}\rangle \leq \langle A_t, \widehat{\theta}_{n_1}\rangle$. This implies

$$
\begin{aligned}
R_\theta(n) &\leq \mathbb{E}_\theta\Big[2n_1 R_{\max} + \sum_{t=n_1+1}^n \langle \theta - \widehat{\theta}_{n_1}, x^* - A_t\rangle\Big] \\
&\leq \mathbb{E}_\theta\Big[2n_1 R_{\max} + \sum_{t=n_1+1}^n \big\|\theta - \widehat{\theta}_{n_1}\big\|_1 \big\|x^* - A_t\big\|_\infty\Big].
\end{aligned}
\tag{B.15}
$$

**Step 2: fast sparse learning.** It remains to bound the estimation error of $\widehat{\theta}_{n_1} - \theta$ in $\ell_1$-norm. Denote the design matrix $X = (A_1, \ldots, A_{n_1})^\top \in \mathbb{R}^{n_1 \times d}$, where $A_1, \ldots, A_{n_1}$ are independently drawn according to sampling distribution $\widehat{\mu}$. To achieve a fast rate, one need to ensure $X$ satisfies restricted eigenvalue condition (Condition A.1 in the appendix). Denote the uncentered empirical covariance matrix $\widehat{\Sigma} = X^\top X / n_1$. It is easy to see

$$
\Sigma = \mathbb{E}(\widehat{\Sigma}) = \int_{x\in\mathcal{A}} xx^\top d\widehat{\mu}(x),
$$

where $\widehat{\mu}$ is the solution of optimization problem Eq. (4.1). To lighten the notation, we write $C_{\min} = C_{\min}(\mathcal{A})$. Since action set $\mathcal{A}$ spans $\mathbb{R}^d$, we know that $\sigma_{\min}(\Sigma) = C_{\min} > 0$. And we also denote $\sigma_{\max}(\Sigma) = C_{\max}$ and the notion of restricted eigenvalue as follows.

**Definition B.1.** Given a symmetric matrix $H \in \mathbb{R}^{d\times d}$ and integer $s \geq 1$, and $L > 0$, the restricted eigenvalue of $H$ is defined as

$$
\phi^2(H, s, L) := \min_{\mathcal{S}\subset[d], |\mathcal{S}|\leq s} \min_{\theta\in\mathbb{R}^d} \Big\{ \frac{\langle \theta, H\theta\rangle}{\|\theta_\mathcal{S}\|_1^2} : \theta \in \mathbb{R}^d, \|\theta_{\mathcal{S}^c}\|_1 \leq L\|\theta_\mathcal{S}\|_1 \Big\}.
$$

It is easy to see $X\Sigma^{-1/2}$ has independent sub-Gaussian rows with sub-Gaussian norm $\|\Sigma^{-1/2} A_1\|_{\psi_2} = C_{\min}^{-1/2}$ (see Vershynin [2010] for a precise definition of sub-Gaussian rows and sub-Gaussian norms). According to Theorem 10 in Javanmard and Montanari [2014] (essentially from Theorem 6 in Rudelson and Zhou [2013]), if the population covariance matrix satisfies the restricted eigenvalue condition, the empirical covariance matrix satisfies it as well with high probability. Specifically, suppose the rounds in the exploration phase satisfies $n_1 \geq 4c_* m C_{\min}^{-2} \log(ed/m)$ for some $c_* \leq 2000$ and $m = 10^4 s C_{\max}^2 / \phi^2(\Sigma, s, 9)$. Then the following holds:

$$\mathbb{P}\Big(\phi(\widehat{\Sigma}, s, 3) \geq \frac{1}{2}\phi(\Sigma, s, 9)\Big) \geq 1 - 2\exp(-n_1/(4c_* C_{\min}^{-1/2})).$$

Noticing that $\phi(\Sigma, s, 9) \geq C_{\min}^{1/2}$, it holds that

$$\mathbb{P}\Big(\phi^2(\widehat{\Sigma}, s, 3) \geq \frac{C_{\min}}{2}\Big) \geq 1 - 2\exp(-c_1 n_1),$$

where $c_1 = 1/(4c^* C_{\min}^{-1/2})$. This guarantees $\widehat{\Sigma}$ satisfies Condition A.1 in the appendix with $\kappa = C_{\min}/2$. It is easy to see Condition A.2 holds automatically. Applying Theorem A.3 in the appendix of the Lasso error bound, it implies:

$$\big\|\widehat{\theta}_{n_1} - \theta^*\big\|_1 \leq \frac{2}{C_{\min}} \sqrt{\frac{2s^2(\log(2d) + \log(n_1))}{n_1}}.$$

with probability at least $1 - \exp(-n_1)$.

**Step 3: optimize the length of exploration.** Define an event $\mathcal{E}$ as follows:

$$\mathcal{E} = \Big\{ \phi(\widehat{\Sigma}, s, 3) \geq \frac{C_{\min}^{1/2}}{2}, \big\|\widehat{\theta}_{n_1} - \theta^*\big\|_1 \leq \frac{2}{C_{\min}} \sqrt{\frac{2s^2(\log(2d) + \log(n_1))}{n_1}} \Big\}.$$

We know that $\mathbb{P}(\mathcal{E}) \geq 1 - 3\exp(-c_1 n_1)$. Note that $\|x^* - A_t\|_\infty \leq 2$. According to Eq. (B.15), we have

$$R_\theta(n) \leq \mathbb{E}_\theta\Big[\Big(2n_1 R_{\max} + \sum_{t=n_1+1}^{n} \big\|\theta - \widehat{\theta}_{n_1}\big\|_1 \big\|x^* - A_t\big\|_\infty\Big) \mathbb{1}(\mathcal{E})\Big] + n R_{\max} \mathbb{P}(\mathcal{E}^c)$$

$$\leq n_1 R_{\max} + (n - n_1)\frac{4}{C_{\min}} \sqrt{\frac{2s^2(\log(2d) + \log(n_1))}{n_1}} 2 + 3n R_{\max} \exp(-c_1 n_1)$$

with probability at least $1 - \delta$. By choosing $n_1 = n^{2/3}(s^2 \log(2d))^{1/3} R_{\max}^{-2/3}(2/C_{\min}^2)^{1/3}$, we have

$$R_n \leq (sn)^{2/3}(\log(2d))^{1/3} R_{\max}^{1/3}\Big(\frac{2}{C_{\min}^2}\Big)^{1/3} + 3n R_{\max} \exp(-c_1 n_1).$$

We end the proof.

### B.4 Proof of Theorem 5.2: improved regret upper bound

We start from a simple regret decomposition based on feature selection step and restricted linear bandits step:

$$R_\theta(n) = \mathbb{E}_\theta\Big[\sum_{t=1}^{n} \big\langle \theta, x^* - A_t \big\rangle\Big]$$

$$= \mathbb{E}_\theta\Big[2n_2 R_{\max} + \sum_{t=n_2+1}^{n} \big\langle \theta, x^* - A_t \big\rangle\Big].$$

**Step 1: sparsity property of Lasso.** We first prove that the Lasso solution is sufficiently sparse. The following proof is mainly from Bickel et al. [2009] with minor changes. To be self-contained, we reproduce it here. Recall that the Lasso estimator in the feature selection stage is defined as

$$\widehat{\theta} = \underset{\theta \in \mathbb{R}^d}{\operatorname{argmin}} \Big(\frac{1}{n_2} \sum_{t=1}^{n_2} \big(Y_t - \langle A_t, \theta \rangle\big)^2 + \lambda_2 \|\theta\|_1\Big).$$

Define random variables $V_j = \frac{1}{n_2} \sum_{t=1}^{n_2} A_{tj}\eta_t$ for $j \in [d]$ and $\eta_t$ is the noise. Since $\|A_t\|_\infty \le 1$, standard Hoeffding's inequality (Proposition 5.10 in Vershynin [2010]) implies

$$\mathbb{P}\Big(|\sum_{t=1}^{n_2} A_{tj}\eta_t| \ge \varepsilon\Big) \le \exp\Big(-\frac{\varepsilon^2}{2n_2}\Big).$$

Define an event $\mathcal{E}$ as

$$\mathcal{E} = \bigcup_{j=1}^{d}\Big\{|V_j| \le \sqrt{\frac{4\log(d)}{n_2}}\Big\}.$$

Using an union bound, we have

$$\mathbb{P}(\mathcal{E}^c) \le 1/d.$$

From the Karush–Kuhn–Tucker (KKT) condition, the solution $\widehat{\theta}$ satisfies

$$\frac{1}{n_2}\sum_{t=1}^{n_2} A_{tj}^\top(Y_t - A_t^\top\widehat{\theta}) = \lambda_2 \mathrm{sign}(\widehat{\theta}_j), \text{ if } \widehat{\theta}_j \ne 0;$$

$$\Big|\frac{1}{n_2}\sum_{t=1}^{n_2} A_{tj}^\top(Y_t - A_t^\top\widehat{\theta})\Big| \le \lambda_2, \text{ if } \widehat{\theta}_j = 0.$$

(B.16)

Therefore,

$$\frac{1}{n_2}\sum_{t=1}^{n_2} A_{tj}(A_t^\top\theta - A_t^\top\widehat{\theta}) = \frac{1}{n_2}\sum_{i=1}^{n_2} A_{tj}(Y_t - A_t^\top\widehat{\theta}) - \frac{1}{n_2}\sum_{i=1}^{n_2} A_{tj}\eta_t$$

Since $\lambda_2 = 4\sqrt{\log(d)/n_2}$, under event $\mathcal{E}$, we have

$$\Big|\frac{1}{n_2}\sum_{t=1}^{n_2} A_{tj}(A_t^\top\theta - A_t^\top\widehat{\theta})\Big| \ge \lambda_2/2, \text{ if } \widehat{\theta}_j \ne 0.$$

And

$$\frac{1}{n_2^2}\sum_{j=1}^{d}\Big(\sum_{t=1}^{n_2} A_{tj}(A_t^\top\theta - A_t^\top\widehat{\theta})\Big)^2 \ge \sum_{j:\widehat{\theta}_j \ne 0}\Big(\frac{1}{n_2}\sum_{t=1}^{n_2} A_{tj}(A_t^\top\theta - A_t^\top\widehat{\theta})\Big)^2$$

$$\ge |\mathrm{supp}(\widehat{\theta}_{n_2})|\lambda_2^2/4.$$

On the other hand, let $X = (A_1,\dots,A_{n_2})^\top \in \mathbb{R}^{n_2 \times d}$ and $\phi_{\max} = \sigma_{\max}(XX^\top/n_2)$. Then we have

$$\frac{1}{n_2^2}\sum_{j=1}^{d}\Big(\sum_{t=1}^{n_2} A_{tj}\Big(A_t^\top\theta - A_t^\top\widehat{\theta}\Big)\Big)^2$$

$$= \frac{1}{n_2^2}\Big(X\theta - X\widehat{\theta}\Big)^\top XX^\top\Big(X\theta - X\widehat{\theta}\Big) \le \phi_{\max}\frac{1}{n_2}\|X\widehat{\theta} - X\theta\|_2^2.$$

Therefore, with probability at least $1 - 1/p$,

$$|\mathrm{supp}(\widehat{\theta}_{n_2})| \le \frac{4\phi_{\max}}{\lambda_2^2 n_2}\|X\widehat{\theta} - X\theta\|_2^2.$$

(B.17)

To lighten the notation, we write $C_{\min} = C_{\min}(\mathcal{A})$. As proven in Section B.3, $X^\top X/n_2$ satisfies Condition A.1 with $\kappa = C_{\min}/2$ when $n_2 \gtrsim s\log(d)$. Applying the in-sample prediction error bound in Theorem A.3, we have with probability at least $1 - 1/p$,

$$\frac{1}{n_2}\big\|X\widehat{\theta} - X\theta\big\|_2^2 \le \frac{9}{C_{\min}}\frac{s\log(d)}{n_2}.$$

(B.18)

Putting Eqs. (B.17) and (B.18) together, we have with probability at least $1 - 2/d$.

$$|\mathrm{supp}(\widehat{\theta})| \le \frac{9\phi_{\max}s}{C_{\min}}.$$

(B.19)

**Step 2: variable screening property of Lasso.** Under Condition 5.1 and using Theorem A.3, it holds that with probability at least $1 - 1/d$,

$$\min_{j \in \text{supp}(\theta)} |\theta_j| > \left\| \widehat{\theta} - \theta \right\|_2 \geq \left\| \widehat{\theta} - \theta \right\|_\infty.$$

If there is a $j \in \text{supp}(\theta)$ but $j \notin \text{supp}(\widehat{\theta})$, we have

$$|\widehat{\theta}_j - \theta_j| = |\theta_j| > \left\| \widehat{\theta} - \theta \right\|_\infty.$$

On the other hand,

$$|\widehat{\theta}_j - \theta_j| \leq \left\| \widehat{\theta} - \theta \right\|_\infty,$$

which leads a contradiction. Now we conclude that $\text{supp}(\widehat{\theta}) \supseteq \text{supp}(\theta)$. We reproduce Theorem 22.1 in Lattimore and Szepesvári [2020] for the regret bound of phase elimination algorithm for stochastic linear bandits with finitely-many arms.

**Theorem B.2.** The $n$-steps regret of phase elimination algorithm satisfies

$$R_n \leq C \sqrt{nd \log(Kn)},$$

for an appropriately chosen universal constant $C > 0$.

Together with Eq. (B.19), we argue the regret of running phase elimination algorithm (Section 22 in Lattimore and Szepesvári [2020]) on $\text{supp}(\widehat{\theta})$ for the rest $n - n_2$ rounds can be upper bounded by

$$\mathbb{E}_\theta \Big[ \sum_{t=n_2+1}^{n} \langle \theta, x^* - A_t \rangle \Big] \leq C \sqrt{\frac{9\phi_{\max}}{C_{\min}} s(n - n_2) \log(K(n - n_2))}.$$

This ends the proof.

## C   Supporting lemmas

**Lemma C.1** (Bretagnolle-Huber inequality). Let $\mathbb{P}$ and $\widetilde{\mathbb{P}}$ be two probability measures on the same measurable space $(\Omega, \mathcal{F})$. Then for any event $\mathcal{D} \in \mathcal{F}$,

$$\mathbb{P}(\mathcal{D}) + \widetilde{\mathbb{P}}(\mathcal{D}^c) \geq \frac{1}{2} \exp\left( -\text{KL}(\mathbb{P}, \widetilde{\mathbb{P}}) \right), \tag{C.1}$$

where $\mathcal{D}^c$ is the complement event of $\mathcal{D}$ $(\mathcal{D}^c = \Omega \setminus \mathcal{D})$ and $\text{KL}(\mathbb{P}, \widetilde{\mathbb{P}})$ is the KL divergence between $\mathbb{P}$ and $\widetilde{\mathbb{P}}$, which is defined as $+\infty$, if $\mathbb{P}$ is not absolutely continuous with respect to $\widetilde{\mathbb{P}}$, and is $\int_\Omega d\mathbb{P}(\omega) \log \frac{d\mathbb{P}}{d\widetilde{\mathbb{P}}}(\omega)$ otherwise.

The proof can be found in the book of Tsybakov [2008]. When $\text{KL}(\mathbb{P}, \widetilde{\mathbb{P}})$ is small, we may expect the probability measure $\mathbb{P}$ is close to the probability measure $\widetilde{\mathbb{P}}$. Note that $\mathbb{P}(\mathcal{D}) + \mathbb{P}(\mathcal{D}^c) = 1$. If $\widetilde{\mathbb{P}}$ is close to $\mathbb{P}$, we may expect $\mathbb{P}(\mathcal{D}) + \widetilde{\mathbb{P}}(\mathcal{D}^c)$ to be large.

**Lemma C.2** (Divergence decomposition). Let $\mathbb{P}$ and $\widetilde{\mathbb{P}}$ be two probability measures on the sequence $(A_1, Y_1, \ldots, A_n, Y_n)$ for a fixed bandit policy $\pi$ interacting with a linear contextual bandit with standard Gaussian noise and parameters $\theta$ and $\widetilde{\theta}$ respectively. Then the KL divergence of $\mathbb{P}$ and $\widetilde{\mathbb{P}}$ can be computed exactly and is given by

$$\text{KL}(\mathbb{P}, \widetilde{\mathbb{P}}) = \frac{1}{2} \sum_{x \in \mathcal{A}} \mathbb{E}[T_x(n)] \langle x, \theta - \widetilde{\theta} \rangle^2, \tag{C.2}$$

where $\mathbb{E}$ is the expectation operator induced by $\mathbb{P}$.

This lemma appeared as Lemma 15.1 in the book of Lattimore and Szepesvári [2020], where the reader can also find the proof.