[Reviews · NeurIPS 2020]

Review 1

Summary and Contributions: Authors give a minimax regret lower bound (of \Sigma(n^(2/3)))which is dimension-free for sparse linear bandits in the data-poor regime where the horizon is larger than the ambient dimension and where the feature vectors admit a well-conditioned exploration distribution. Authors also give matching upper bound and with further assumption give a stronger upper bound (of O(n^(1/2)).

Strengths: 1) Authors clearly point out assumptions of previous work and current work. 2) Bound proved in the paper doesn’t depend on d.

Weaknesses: 1) No empirical evidence that proposed methods work. Even synthetic data experiments in the data regime proposed could have been enough to convince the importance of the paper. 2) Algorithm 1 - Explore the sparsity then commit (ESTC) has two phases. First phase looks like a pure exploration phase and second phase looks like a pure exploitation phase. How does this algorithm compare to existing similar algorithm like - [1] 3) Cmin - the minimum eigenvalue of the data matrix for an exploration distribution. Is it possible to improve dependency on Cmin in upper bound or lower bound? Can authors comment more on this? [1] Deshmukh, Aniket Anand, et al. "Simple regret minimization for contextual bandits." arXiv preprint arXiv:1810.07371 (2018). I am satisfied with the rebuttal and experimental simulation and I have updated my score.

Correctness: Looks correct!

Clarity: Well written,

Relation to Prior Work: Yes.

Reproducibility: Yes

Additional Feedback:


Review 2

Summary and Contributions: The paper studies the stochastic linear bandits with high dimensional sparse features. For this problem, the paper establishes a novel n^0.66 dimension free minimax regret lower bound and complement it by an explore-then-exploit algorithm that nearly matches the established lower bound.

Strengths: The paper provides a thorough investigation of the high dimensional sparse linear bandits and presents novel results that should be of interest to wide audience given the popularity of linear bandits both in theory and practice. In particular the dimension free lower bound of n^0.66 is an improvement over existing linear regret bounds for data poor regimes (i.e. dimension of feature space > time horizon). The paper also presents fairly simple but impactful algorithms that explore the sparsity and then commit (exploit) whose performance nearly matches the established lower bounds. Some of the algorithmic design aspects like sampling from the design distribution are new for this problem setting and could be of relevance for other sparse bandit problems (for eg. sparse high dimensional generalized linear bandits). The later could be follow up work of this paper.

Weaknesses: The paper could do a better job of clarifying/remarking some of the technical claims/discussions. (see comments below). Post rebuttal: I acknowledge that I have read and taken other reviewers feedback into account the rebuttal. Based on the discussion, I am decreasing the score by 1.

Correctness: Yes

Clarity: The paper is very well written, though there is some room for improvement in the technical discussion.

Relation to Prior Work: The paper does a good job of relating to existing literature and the contributions.

Reproducibility: Yes

Additional Feedback: 1. One of my concern is that there's a potential chance for mis-interpreting some of the claims made by the paper. For example, the paper claims that the established lower bound is dimension free and is better than existing linear regret for data poor regimes. However as remarked by the authors, the C_min(A) could be as bad d, in which case the regret could be linear (in settings where d > n). This makes the lower bound both dimension dependent and linear in n. The paper should have disclaimer in the abstract or the contribution sections instead of pushing till end of the technical discussion (remark 4.5) 2. I would also like the paper to discuss the relation between the minimum eigen value (and the corresponding distribution) computed by equation 4.1 and sparsity. 3. Lastly, if there are arms that are clearly not optimal but are responsible in decreasing the value of C_min(A), is there an easy way to weed them out? In similar context, what can be done when A doesn't span R^d?


Review 3

Summary and Contributions: This paper studies the linear bundit problem with high dimensional features. A minimax lower bound for the regret is derived at the rate of n^(2/3) in the so-called data-poor regime. A matching upper bound is derived in the same regime. Lastly, with additional assumptions, an improved bound of n^(1/2) is proved. The paper borrows techniques from high dimensional statistics. ==== update: I appreciate the clarification and the efforts to add a few simulation studies. I have bumped my rating by 1.

Strengths: The paper is well written. The difference with other previous results was well articulated.

Weaknesses: 1. I think the "intriguing" transition between n^2/3 and n^1/2 is an overstatement. After all, if d grows with the sample size n, then it has to be factored in. d should not be viewed as a fixed constant. Similarly, the signal level s could also be a function of d, and hence, a function of n. 2. It is interesting that in the data rich regime the author can have a regret lower bound sqrt(dn) that is sqrt(s) smaller than the standard sqrt(sdn) lower bound. Note that s may also grow with n and should not be ignored. I am not fully convinced by the tightness of this lower bound. 3. It is not clear how the optimization (4.1) can be done numerically, if x follows absolutely continuous distribution. Some discussion will be helpful. 4. Much of the proofs can be moved to the supplementary materials to make rooms for some numerical studies.

Correctness: The claims seem alright, despite being overstated at places.

Clarity: Yes.

Relation to Prior Work: Yes.

Reproducibility: No

Additional Feedback:

[Author Response · NeurIPS 2020]

We thank the reviewers for their valuable comments! We have added comprehensive empirical studies and hope you are
satisfied with our point-by-point responses and increase your scores!

**(Experiments):** We agree that adding experiments is a good idea and have completed an extensive empirical evaluation.
Given the space limitation in the response, only a subset is included below. All the experiments will be added in the
revised version. We compare ESTC (our algorithm) with LinUCB [2] and doubly-robust (DR) lasso bandits [1]. For
ESTC, we use the theoretically suggested length of exploration stage. For LinUCB, we use the theoretically suggested
confidence interval. For DR-lasso, we use the code made available by the authors on-line.

**Case 1: linear contextual bandits.** We use the setting in Section 5 of [1] with $N = 20$ arms, dimension $d = 100$,
sparsity $s = 5$. At round $t$, we generate the action set from $N(0_N, V)$, where $V_{ii} = 1$ and $V_{ik} = \rho^2$ for every $i \neq k$.
Larger $\rho$ corresponds to high correlation setting that is more favorable to DR-lasso. The noise is from $N(0,1)$ and
$\|\theta\|_0 = s$. **Case 2: hard problem instance.** Consider the hard problem instance in the proof of minimax lower bound
(Thm 3.3), including an informative action set and an uninformative action set.

**Conclusion:** The experiments confirm our theoretical findings. Although our theory focuses on the fixed action set
setting, ESTC works well in the contextual setting. DR-lasso bandits heavily rely on context distribution assumption
and almost fail for the hard instance. LinUCB suffers in the data-poor regime since it ignores the sparsity information.
We do not evaluate [3] since it is not a polynomial-time algorithm.

Figure 1: The left two figures are for Case 1 and the right two figures are for Case 2.

**Reviewer #1. (Compare with [4]):** Thanks the reference, which will be included in a revised version. The algorithm
in [4] and ESTC share the explore-then-commit template but both the exploration and exploitation stages are very
different. [4] considers simple regret minimization while we focus on cumulative regret minimization. **(Dependence**
**on $C_{\min}$):** Surprisingly, even in the classical statistical settings there are still gaps between upper and lower bounds. We
speculate that the upper bound may be improvable, though at present we do not know how to do it. A discussion will be
included in the revised version.

**Reviewer #2. (Interpreting of claims):** We agree with this comment and will make this clear up-front. **(Relation**
**between eigenvalue and sparsity):** The optimization problem in Eq. (4.1) only depends on the action set and not the
sparsity.

**(Weeding out actions):** This is an interesting question. As the second part of your question hints, things are already
delicate when the actions do not span $\mathbb{R}^d$. One cannot expect the Lasso estimate to be close to the true parameter because
there is no information in some directions. We are not aware of direction-dependent confidence bounds for Lasso that
are suitable in this case. A standard idea in non-sparse settings is to change the coordinates to the low-dimensional
subspace, but rotations do not preserve sparsity, so this does not work here. For the first part of your question. A small
value of $C_{\min}$ sometimes happens when most actions pointing in some direction are quite short. You might hope to
learn that these actions cannot be optimal and then work in the low dimensional subspace, so solving one problem may
help with the other.

**Reviewer #4. (Transition between $n^{2/3}$ and $n^{1/2}$):** Our bounds are non-asymptotic and our intention is not to treat
any quantities as constants. The bounds show that there is a rich information-tradeoff in sparse linear bandits that
appears in the high-dimensional regime. In particular, for certain action sets algorithms can enjoy nearly dimension-free
regret by exploring carefully while algorithms based on optimism may be very suboptimal.

**(Tightness of lower bound in the data-rich regime):** We do not claim our lower bound is tight in the data-rich regime
$(d < n)$ where a lower bound of $\Omega(\sqrt{dsn})$ is already known to be optimal. **(Solving the optimization problem):**
When the number of arms is finite it can be solved using standard convex solvers since the minimum eigenvalue is a
concave function. If the number of arms is infinite, things will likely be delicate in general. Hints may be found in the
literature on optimal design where computational questions remain open.

[1]. Doubly-robust lasso bandit. NeurIPS 2019. [2]. Improved algorithms for linear stochastic bandits. NIPS 2011. [3].
Online-to-confidence-set conversions and application to sparse stochastic bandits. AISTATS 2012. [4]. Simple regret
minimization for contextual bandits. ArXiv 2018.


[Meta-Review · NeurIPS 2020]

This papers views high dimensional sparse bandits from an interesting data-poor angle. The discussion of lower bounds in this setting versus the standard setting is very interesting. It is interesting that the rate becomes n^{2/3} in the data poor regime and that it is matched by an explore-exploit algorithm.